# Status, Sources and Assessment of Potentially Toxic Element (PTE) Contamination in Roadside Orchard Soils of Gaziantep (Türkiye)

**DOI:** 10.3390/ijerph20032467

**Published:** 2023-01-30

**Authors:** Mustafa Demir, Erdihan Tunç, Sören Thiele-Bruhn, Ömer Çelik, Awet Tekeste Tsegai, Nevzat Aslan, Sevgi Arslan

**Affiliations:** 1Biology Department, Faculty of Art and Sciences, Gaziantep University, University Boulevard, Şehitkamil, 27310 Gaziantep, Türkiye; 2Soil Science Department, Regional and Environmental Sciences, Trier University, Campus II, D-54286 Trier, Germany; 3Department of Plant Production, Faculty of Applied Sciences, Muş Alpaslan University, 49250 Muş, Türkiye; 4Independent Researcher, 1696 Ball Avenue, Grand Rapids, MI 49505, USA; 5Pistachio Research Institute, Ministry of Agricultural and Forestry, University Boulevard No: 31, 27060 Gaziantep, Türkiye

**Keywords:** soil, PTE, pollution assessment, pistachio, olive, *SOPI*

## Abstract

To identify the sources of contamination with potentially toxic elements (PTEs) in roadside orchard soils and to evaluate the potential ecological and environmental impacts in Gaziantep, soil samples from 20 mixed pistachio and olive orchards on roadsides with different traffic densities and at different distances to the roads were analyzed. Concentrations were 23,407.36 ± 4183.76 mg·kg^−1^ for Fe, 421.78 ± 100.26 mg·kg^−1^ for Mn, 100.20 ± 41.92 mg·kg^−1^ for Ni, 73.30 ± 25.58 mg·kg^−1^ for Cr, 65.03 ± 12.19 mg·kg^−1^ for Zn, 60.38± 7.91 mg·kg^−1^ for Pb, 17.74 ± 3.35 mg·kg^−1^ for Cu, 14.93 ± 4.94 mg·kg^−1^ for Co, and 0.30 ± 0.12 mg·kg^−1^ for Cd. It was found that the Ni content in 51% and the Cr content in 18% of orchard soils were above the legal limits for agricultural soils (pH > 6) in Türkiye. Factor analysis (FA) showed that Co, Cr, Cu, Fe, Mn, Ni, and Pb loaded on the first factor (FC1), while Cd and Zn loaded mostly on the second factor (FC2). It was found that Cr, Ni, and Pb were primarily enriched through pedogenic processes, whereas Cd most likely originated from agricultural activities, while the impact of road traffic as source of PTE contamination was insignificant. It has been revealed that the soils are of low quality for agricultural production due to PTE contamination (*PIave* ≥ 1). The *SOPI* values from environmental and ecological individual indices showed that the soil pollution level was moderate for Cd, Ni, and Pb, and low for Cr. The soil pollution index (*SOPI*) proved to be suitable for evaluating and comparing PTE pollution in regions with different soil properties.

## 1. Introduction

Some heavy metals (HMs) with densities greater than 5 g/cm^3^ have a toxic or ecotoxic capacity depending on their concentration [1] and, thus, are termed as potentially toxic elements (PTEs). These PTEs are frequently found as soil contaminants [2,3]. Because PTEs are inorganic pollutants and, thus, are not biodegradable, they persist in soil and can be further transferred to crop plants and through the food chain to animals and humans [4,5]. Some PTEs, such as Zn and Cu, are nutrients for living organisms and essential for the organisms’ metabolic activities [6,7]. However, it is a great concern that PTEs can accumulate in soil, organisms, and humans so that they occur at concentrations above toxicity levels [8,9]. The accumulation of PTEs in soils first of all results from the weathering of geogenic mineral resources; however, since the industrial revolution, PTE accumulation from additional anthropogenic sources largely exceeds the accumulation based on natural processes [10,11]. The main anthropogenic sources from which PTEs can originate are mining, industrial and urban solid wastes and emissions, and agricultural use of PTE-containing pesticides and fertilizers, as well as emissions and wastes from transportation [12,13]. Traffic emissions comprise, among other environmentally relevant compounds, PTEs, such as Cd, Co, Cr, Cu, Ni, Pb, and Zn. Hence, roadside soils are, on the one hand, exposed to ongoing contamination and, on the other hand, different to soils in industrial or mining areas, typically used for agriculture. This makes a possible pollution especially relevant [14]. Sources of PTEs at roadside are generally divided into five groups, i.e., emissions from fuel combustion by traffic and cargo, abrasion of pavement and filling material, road equipment, maintenance and operation, and external resources [2,15]. These sources can cause contamination in natural and agricultural soils, especially along roadsides [16,17]. A wide array of elements, i.e., Ag, As, Cd, Co, Cr, Cu, Mn, Mo, Ni, Pb, Pd, Pt, Rh, Sb, Si, Ti, and Zn are released from the engine parts, brake lining systems, tires or metal parts of the vehicles, and through exhaust emissions that result from the use of fuel and motor oil [2,15].

Traffic density (TrD) and distance to the road (DtR) have important effects on the concentrations of PTEs in soils along roadsides. The PTE concentrations in roadside soils often increase with an increase in traffic density [6,17]. Previous studies showed that PTE pollution can be found at distances between 10 m and 25 m from the road [18,19]. Even larger distances of 100 m and more have been suggested for soil sampling in studies on PTEs [2,17] because the spatial distributions of traffic-based PTE contamination varies with factors, such as soil particle size, wind, precipitation, and vegetation [16,20]. In contrast, a decrease in PTE concentrations with distance to the road in agricultural soils cannot be assigned to agricultural practices, such as fertilization, tillage and irrigation [16].

Soil pH, organic matter (SOM), clay, and carbonates (IC) are relevant factors that determine the mobility of PTEs in soil [5,21,22]. Complexes formed with SOM and clay serve as an important sink for PTEs in soil [23,24]. In addition, high pH associated with carbonate in soils increases the immobilization of PTEs in the presence of soil colloids [10,23,25]. 

Avoiding PTE contamination of agricultural soils is important not only for the protection of soil and environmental quality and health but also for human health by avoiding contamination of crops and fodder plants [11,26]. In this respect, determining the degree and sources of PTE pollution in agricultural soils is important. While multivariate statistical analyses [20,27] and spatial distribution maps [12,28,29] were used to identify PTE sources, various individual or complex indices were used to determine PTE pollution levels [9,30]. In general, the levels of PTE pollution were determined by using the PTE concentrations of the studied area, in comparison with pre-industrial levels or reference geochemical background values, e.g., by calculating environmental and ecological assessment indices. Eighteen different assessment indices, such as the contamination factor (*Cf*), are described in the literature [31]. However, none of these indices considers the contents of soil constituents, such as SOM, IC, or clay, or soil properties, such as pH, which are known to have significant effects on the accumulation of PTEs in soils [20,32]. 

Due to the threat posed to human life and nature, PTE contamination in agricultural areas attracted the attention of scientists many years ago [33,34]. Worldwide, many studies have been carried out to determine the contamination and sources of PTEs in agricultural areas [10,22,26]. Similar studies have been carried out to evaluate PTE pollution in agricultural areas and to determine its sources in Türkiye [11,29,35]; however, there are no studies on the ecological and environmental assessment and the determination of sources of PTE pollution in the agricultural areas of Gaziantep, Anatolia. Vehicle traffic has increased rapidly in Gaziantep in the last decades due to industrial developments and rapidly increasing population growth especially contributed by post-war Syrian immigrants. This raises questions about the contamination of agricultural soils in such an environment of strongly intensifying anthropogenic influence. Answering these questions will reveal the universally important and basic information about maintaining the productivity and health of these roadside orchard soils where pistachio and olive agriculture is economically important and where dry farming is practiced. Hence, the aims of this study were (i) to determine PTE contents and contamination levels in roadside orchards, (ii) to identify PTE contamination sources, (iii) to determine the relationship of PTE contamination with traffic density and distance from the road, and (iv) to create a new index in which PTE accumulation can be evaluated according to soil general properties. This was carried out for a set of nine selected metals, representing relevant PTEs from traffic (Cd, Pb, Cr, Co, Ni) and/or from agricultural origin (Cu, Zn), or metals that are typical soil constituents (Fe, Mn).

## 2. Materials and Methods

### 2.1. Study Area and Sampling

The study area with geographical coordinates is shown in the map in Figure 1. Gaziantep, located in the southeast of Türkiye, adjacent to Syria, at the point connecting the Southeastern Anatolia Region to the Mediterranean and Central Anatolia Regions and on the historical Silk Road, is an important industrial and agricultural city and its population is increasing rapidly. In this respect, transportation is important for Gaziantep and it is increasing with increasing industrial activities and population. The D400 highway stretches from west to east, connecting Gaziantep to Adana in the west and to Şanlıurfa in the east. The D850 highway, which is in the north–south direction, connects the city to Kilis in the south and Adıyaman in the north (Figure 1). Soil samples were taken at selected orchard sites along these highways. According to the vehicle counts carried out at the stations since 2004 (Appendix A), the average vehicle traffic on the Gaziantep–Adana highway (GAH; D400), Gaziantep–Kilis highway (GKH; D850), Gaziantep–Nizip highway (GNH; D400), and Gaziantep–Yavuzeli highway (GYH; D850) was 245, 225, 479, and 180 vehicles.h^−1^, respectively [36]. Therefore, according to the number of vehicles per hour, GYH (<200 vehicles.h^−1^) is considered to be a low traffic density road (LTD), GAH and GKH (200 < vehicles.h^−1^ < 400) are medium traffic density roads (MTD), and GNH (vehicles.h^−1^ > 400) is considered as a high traffic density road (HTD).

A total of 80 soil samples were taken from a total of 20 different pistachio and olive orchards (sampling areas) at a distance of at least 2 km from each orchard (sampling localities), from a distance of 10, 25, 50, and 100 m from the roads (sampling points), from a depth of 0–20 cm with a stainless-steel shovel. Samples were collected in September 2019. The selected orchards had the same elevation as the road and were far from industrial (>5 km) and residential areas (>2 km). A total of 80 soil samples, each with 5 replicates, were taken from the surface at a depth of 0–20 cm at 10 m, 25 m, 50 m, and 100 m distances (sampling points) from the roads (Figure 1). 

### 2.2. Preparation and Analysis of Soil Samples

Soil samples were further processed at the Gaziantep University Araban Vocational School, Soil Analysis, Research and Application Center laboratories (Gaziantep, Türkiye). Before analysis, soil samples were air-dried at room temperature and subsequently sieved < 2 mm. Five replicate soil samples taken from each sampling point were weighed in equal weights, before being combined and homogenized by mixing to prepare composite samples. All the glassware that was used for the analysis of soil samples was kept in nitric acid solution (HNO_3_/H_2_O, 1:1 *v*/*v*) for 24 h to prevent contamination. It was then washed with 1% HNO_3_ solution and dried in an oven (Nüve, FN 055, Ankara, Türkiye) at 50 °C. 

Soil pH was determined according to Jones [37] in a 1:2.5 (*w*/*w*) soil–water mixture with the aid of a multi-parameter analyzer (Consort C5020, Turnhout, Belgium). The inorganic carbon content (IC) from carbonates in the soils was determined using a Scheibler calcimeter [38]. The Walkley–Black titration method was used to determine the SOC (%) of the soil samples [39]. The SOM (%) content was calculated as SOC (%) × 1.724 [40]. The silt, clay, and sand contents of the soils were determined in a Bouyoucos hydrometer according to the method based on measuring the density of the suspension after dispersion [41]. The PTE analyses were performed at the Eastern Mediterranean Transition Region Agricultural Research Institute (Kahramanmaraş, Türkiye). Soil samples and reference material (UME EnvCRM03, TÜBİTAK, Kocaeli, Türkiye) were extracted with aqua regia. A total of 0.25 g of soil sample was placed in Teflon tubes, and 10 mL of a mixture of HNO_3_ (70%, Merck, Darmstadt, Germany) and HCl (37%, Sigma, Buchs, Switzerland) was added (1:3 v/v). The extraction process was carried out in a microwave oven (CEM Mars 5 Express, Matthews, NC, USA), with ramp and hold cycles of 15 min at 1200 W at 200 °C. Samples were subsequently filtered with 10% HCl using a Whatman No 42 filter to have a final volume of 25 mL. The PTE concentrations in the extracts were determined using 5100 ICP-OES (Agilent Technologies, Mulgrave, Australia) according to Olesik [42]. Five replicates were made for the reference material (Appendix A). Furthermore, the quantification limits (LOQ) and detection limits (LOD) for the metals and PTEs based on SD values of 10 individual blank samples are given in Appendix A. All analyses of soil samples were performed in triplicate, and the results were combined by calculating the arithmetic mean. Analyses were repeated in case the standard deviation between replicates exceeded 5%.

### 2.3. Evaluation of Environmental and Ecological Risks

Individual and complex indices proposed by Kowalska et al. [31] for the environmental and ecological assessment of PTE contamination in agricultural soils were also used in this study. These indices refer to individual or the sum of all PTEs tested, without or with reference to global and/or local background levels of metals, and in some cases include toxicity measures or even soil physicochemical parameters. The ranges and classifications of environmental and ecological indices are given in Appendix A.

One of the individual indices frequently used in the assessment of soil pollution is the contamination factor (*Cf*) introduced by Håkanson [43]. The *Cf* is calculated using the following equation:(1)$$Cf_{n}=\frac{C_{n}}{C_{PIV}}$$
where *C_n_* is the mean concentration of the PTE (mg·kg^−1^) nth PTE, and *C_PIV_* is the pre-industrial reference value (PIV, Appendix A) of the nth PTE recommended by Håkanson [43].

With the modified degree of contamination (*mCd*) suggested by Abrahim and Parker [44], it makes it possible to evaluate the contamination of the research site for all studied PTEs. Thus, *mCd* is calculated using the following equation: (2)mCd=∑i=1nCfn
where *n* is the number of PTEs whose *Cf* value is detected.

The potential ecological risk factor (*ER*) and potential ecological risk index (*RI*) proposed by Håkanson [43] are calculated with the help of the following equations: (3)$$ER_{n}=TR_{n}\times Cf_{n}$$
(4)$$RI=\overset{n}{\underset{i=1}{\sum }}ER$$
where *Cf_n_* is the value of the contamination factor nth PTE, and *TR_n_* is the toxic response factor for nth PTE (Appendix A). 

The single pollution index (*PI*) is frequently used for the assessment of individual metals or PTEs in topsoil [45,46]. In *PI*, calculated similarly to *Cf*, local or reference geochemical background values (RGB) are used instead of PIV. As such, *PI* is calculated using the following equation:(5)$$PI=\frac{C_{n}}{CGB_{n}}$$
where *C_n_* is the mean concentration of the nth metal or PTE studied (mg·kg^−1^) and *CGB_n_* is the mean of upper continental crust contents (UCC, mg·kg^−1^) of the nth metal or PTE. Although it is recommended to use both local and reference geochemical background content [31], since there are no data on the local geochemical background contents of the Gaziantep region in the literature, in this study, UCC [47] represents the lithogenic composition of metals as RGB. 

The enrichment factor (*EF*), which is an effective tool used for the evaluation of individual PTE pollution in soils, was proposed by Buat-Menard and Chesselet [48]. The enrichment factor (*EF*) is based on standardizing the PTE concentrations detected in the study with concentrations of a lithophilic reference element. Thus, PTE variability is reduced. The *EF* is calculated using the following equation:(6)EF=(CSnCSREF)(CGBnCGBREF)
where *CS_n_* is the mean concentration (mg·kg^−1^) of the nth metals or PTEs in the studied area, *CS_REF_* is the mean concentration (mg·kg^−1^) of the reference element in the studied area, *CGB_n_* is the mean concentration (mg·kg^−1^) of the nth metals or PTEs in the UCC, and *CGB_REF_* is the mean concentration of UCC (mg·kg^−1^) of the metal accepted as a reference. The most common reference elements used in this index are aluminum (Al), calcium (Ca), Fe, Mn, titanium (Ti), and scandium (Sc). In our study, Fe was used as a reference element, which is expected to be relatively less affected by anthropogenic activities due to its higher concentrations in nature than other reference elements [49]. 

One of the complex indices, the average single pollution index (*PIave*) is the average of the *PI* values of all studied metals and PTEs and is calculated with the following equation [31,45]:(7)PIave=∑n=11PIn
where *n* is the number of metals and PTEs whose *PI* value is detected.

It has been observed that in all indices used to evaluate the pollution caused by PTEs in the soil [31], pH, SOM, IC, and clay contents [23], which have important effects on the accumulation and pollution of PTEs in the soil, are not taken into account. Hence, we propose a new index here. The soil pollution index (*SOPI*) has been developed to determine the degree and source of PTE accumulation in the soil by standardizing the soil with its selected soil general properties. This index can be altered by normalizing PTE concentrations to those of non-toxic, lithogenic elements (Fe and Mn in this study) as in EF. Equations (8) and (9) are as follows:(8)SOPIn=PIn(∑i4Tsp)/4
(9)SOPI=SOPIn(∑inSOPILTG)/nLTG
where *SOPI_n_* is the soil pollution index value determined for nth metals or PTE, *PI_n_* is the *PI* value of the nth metals or PTE, *Tsp* is the mean contents of soil general properties, 4 is the number of soil general properties (pH, SOM, IC, clay) that affect PTE concentrations, *n_Tsp_* is the number of overall soil properties studied, and *n_LTG_* is the number of lithophilic elements studied.

### 2.4. Statistical Analysis

A normality test was applied to the data set obtained as a result of the analyses. A Pearson correlation test was applied to the parameters with normal distribution for correlation analysis. One-way ANOVA with the post hoc Tukey test was used for variation analysis. The Spearman test for correlation analysis, Kruskal–Wallis test, and two related nonparametric sample tests were applied to parameters with non-normal distributions. The IBM SPSS Statistics software (Version 25.0, SPSS Inc., Chicago, IL, USA) was used for these tests. 

Minitab Statistical Software (Version 19, Minitab Ltd., Coventry, UK) was also used for factor analysis (FA) based on grouping factors with strong correlations defined by the factor matrix after varimax rotation. In the selection of the number of factors, eigenvalues ≥1.0 were accepted according to the Kaiser criteria. Principal component analysis was applied as a factor extraction method. Rotated factor loadings ≥ 0.7 obtained from Varimax rotation were considered as a factor.

## 3. Results and Discussion

### 3.1. Selected General Soil Properties of Orchard Soils along Major Highways

All soils were slightly alkaline with an average pH of 7.90 ± 0.06 (7.80 to 8.01). Accordingly, all soils contained carbonate within a range from 2.51% to 35.35% (mean 18.31 ± 11.39%). The mean SOM in soil samples was found to be low with 0.99 ± 0.41% (0.45% to 1.90%). The averages of soil texture components were 23.79 ± 9.89% for clay, 48.69 ± 10.41% for silt, and 27.52 ± 4.80% for sand (Table 1). Hence, soil textures were classified as loam to silt loam, according to WRB. The results showed that orchard soils along roadsides have high pH and carbonate contents and moderate SOM and clay contents, which corresponds to the findings of previous studies in the area of Gaziantep [50,51]. Generally, Gaziantep is located geologically in a zone with limestone and gypsum [52] and climatically in a transition zone between the Mediterranean and continental climates [53]. Hence, carbonate in soil originates from the parent rock material and is poorly leached from the soil in the dry climate with limited annual precipitation of 400–600 mm [54]. The low SOM content in the soils of the region is probably caused by the dry climatic conditions on the one hand and by agricultural practices on the other. In a study conducted in agricultural areas in the region of Gaziantep, pasture soils were found to have higher SOM content than pistachio orchard soils due to agricultural practices, such as irrigation, tillage, and fertilization [55]. Moreover, the removal of the growing annual herbaceous biomass for burning or use as feed, which is widely applied in mixed pistachio and olive orchards in the region, substantially reduces the entry of organic matter into soils [50]. 

It was determined that the average contents of the selected soil general properties did not have significant differences at different distances from the road (*p* > 0.05; Table 1). However, except for soil pH, the contents of all selected soil general properties differed significantly (*p* < 0.05) depending on the location of the investigated highway, which eventually coincided with the traffic density. It was determined that SOM and clay content in LTD and MTD were higher than in HTD, while IC contents in HTD were higher than those in LTD, which is most likely due to natural preconditions and subsequent pedogenic processes [23]. Correspondingly, it was reported in other studies on roadside soils that soil pH was unaffected but soil organic carbon content was significantly higher at roadsides, yet was not correlated with traffic density [20].

### 3.2. Contents of Metals and PTEs of Orchard Soils

#### 3.2.1. Average Concentration of Metals and PTEs 

The average metal and PTE concentrations decreased in the order Fe > Mn > Ni > Cr > Zn > Pb > Cu > Co > Cd (Table 2). The mean concentrations of the investigated metals and PTEs, except for Cd and Pb, are consistent with the concentrations found in uncontaminated (control) soils of previous studies in the Gaziantep region [56,57]. However, the Cd concentrations were lower than in the aforementioned studies, while the Pb concentrations were higher. 

Comparing the results with the legal limit values for agricultural soils with pH > 6 in Türkiye [58], it was found that 51.76% of soils for Ni and 18.75% of soil for Cr on the roadsides are above that limit (Table 2). The average contents of Cd, Cr, Ni, and Pb detected in this study were noticeably higher (approximately 3, 2, 5.5, and 3.5 times, respectively) than the upper continental crust values [47], indicating that the orchard soils may have been contaminated with these PTEs.

#### 3.2.2. Relationship between Metal and PTE Contents in Roadside Soils and Traffic

Contrary to previous studies [30,61] that showed a decreasing contamination level with increasing distance from the road, it was found in this study that both metal and PTE concentrations were not significantly different (*p* > 0.05) at different distances from the road (Figure 2). This finding may be due to long-term agricultural activities, such as soil tillage, causing homogenization and relocation of topsoil material within a field site [16,62]. Furthermore, with increasing traffic density, the concentrations of Co, Cr, Cu, Fe, Mn, Ni, and Pb had a statistically decreasing trend (*p* < 0.05; Figure 2b–h), while Cd and Zn did not have a significant correlation with traffic density (*p* > 0.05; Figure 2a,i). These findings coincide with the findings of previous studies, which reported that the PTE content of soils also increased depending on the traffic density. This is more a result of their association with selected general soil properties, which have been reported to have significant effects on the abundance and mobility of PTEs in soils [13,23], rather than the contribution of traffic emissions to PTE contents in soils.

### 3.3. Source Assignment of Metals and PTEs in Soils

The results of the FA analysis performed to identify the sources of the metals and PTEs showed that there were two factors with an eigenvalue greater than 1, which explained 84.9% of the total variance (Table 3). The first factor (FC1) explains 64.0% of the total variance (Figure 3a) and includes Co, Cr, Cu, Fe, Mn, Ni, and Pb with positive loading (≥±0.7; Figure 3b). According to the correlation analysis, metals and PTEs have significant positive relationships with each other (*p* < 0.05, Table 4). These relationships, which can be explained by the siderophilic properties of Co, Cr, and Ni, and the chalcophilic properties of Cu and Pb, show that a common source caused the concentrations of the PTEs in the orchards [25,63]. Moreover, their significant relationships with the general characteristics of the soil (*p* < 0.05, Table 2) also indicate that the metal and PTE concentrations resulted from pedogenic processes [25]. As expected, the contents of most elements were closely correlated to SOM and pH, since these, in addition to the parent rock, significantly control the metal and PTE contents of soils [23]. The relationships in FC1 indicate that the metals and PTEs are strongly immobilized in the presence of SOM [24] and carbonates associated with alkaline soil pH [10]. Thus, the relationships between general soil characteristics and metals and PTEs explain the seemingly contradictory results that metal and PTE concentrations were lower in orchards at HTD roads than at roadsides with lower traffic densities (Figure 2b–h); soils at HTD had lower SOM and clay, and higher IC contents than others (Table 1). The FA and correlation analysis results confirm that FC1 is the lithogenic factor, which is in line with previous studies reporting that lithogenic PTEs are typically enriched in agricultural soils [7].

The second factor (FC2), which accounts for 20.9% of the total variance, includes Cd and Zn with high loadings (≥±0.7) (Table 3). It was determined that Cd and Zn did not have significant relationships (*p* > 0.05; Table 4) with the other metals and PTEs loading on FC1. Hence, it is concluded that they also differ from the other PTEs in terms of their source, an interpretation which is in agreement with previous studies [26,64]. This assumption is also confirmed by the relationships of Cd and Zn with the general soil properties that are substantially weaker compared to the metals and PTEs loading on FC1 (Table 4). It suggests that both PTEs in the orchard soils are more likely to originate from anthropogenic sources [35]. Both Cd and Zn can be enriched by various agricultural activities, such as the use of phosphate fertilizers produced from contaminated bedrock [65]. For example, high Cd concentrations of up to 0.4 mg·kg^−1^ in pistachio orchard soils in the city of Kerman, Iran, were caused by the use of phosphate fertilizers [27]. Although Cd and Zn are included in the same factor (Figure 3b), probably due to their chalcophilic properties, no significant correlation (*p* > 0.05; Table 4) was detected between both PTEs. Moreover, according to the FA results, Cd has a negative loading and Zn has a positive loading (Table 3). These findings show that Cd and Zn may differ from each other in terms of their sources. In the Gaziantep region, the use of livestock manure as an organic fertilizer is a common practice in pistachio orchards and may have contributed to the Zn levels. It has been reported that 37–40% of the Zn input to agricultural soils in England and Wales [66] is contributed by the use of manure enriched with Zn that was used as a feed additive [65].

### 3.4. Environmental and Ecological Assessment of PTEs in Soils

According to the average PTE contents, the orchard soils have moderate contamination for *mCd* (3.79 ± 1.45; Figure 4a). The *RI* value (117.56 ± 42.24) according to the average PTE contents shows that the orchard soils have low ecological risk (Figure 4b). However, it has also been reported that agricultural soils in India have a high ecological risk (*RI* = 544) due to the long and intensive use of inorganic fertilizers and pesticides [67]. The *PIave* value (2.14 ± 0.45) shows that the soils are of a low quality due to PTE contamination [68] (Figure 4c).

The average contents of Ni have very high contamination for the *Cf* value (20.04 ± 8.34; Figure 5a) and considerable potential ecological risk for the *ER* value (100.20 ± 41.68; Figure 5b). Moreover, according to the average *Cf* values, moderate contamination was detected in 35.94% of the soils for Cr and 9.38% for Pb. In previous studies [69], it has been reported that Ni and Cd have higher *Cf* and *ER* values than other PTEs in roadside agricultural areas in India due to intensive agricultural activities. Contrary to our study, roadside agricultural soils in Lishui (Chinese) have been reported to have a low potential ecological risk for all PTEs for Ni [28]. 

The *PI* values, which suggested that *PI* values greater than 1 may indicate the existence of anthropogenic sources [70], show that the orchard soils are very strongly polluted for Ni, strong for Pb, moderately for Cd and Cr, and that pollution is low in terms of Co, Cu, and Zn (*PI* > 1; Figure 5c). It has been determined that the order of *EF* and *SOPI* values of PTEs is Ni > Pb > Cd > Cr > Zn > Co > Cu > Mn > Fe. According to the *EF* values, Ni (6.76 ± 1.85) was found to be significantly enriched, while Cd, Cr, and Pb (3.96 ± 1.67, 2.66 ± 0.54 and 4.71 ± 0.62, respectively) were moderately enriched in this orchard soil (Figure 5d). In addition, it was found that all of the soils were enriched for Ni and Pb, while for Cd, Co, Cr, and Zn, 78.13%, 20.31%, 98.44%, and 10.94% of the soils were enriched, respectively (*EF* > 2). Similarly, the enrichment of Cr, Ni, and Pb in agricultural soils has been reported due to agricultural practices [12]. The *SOPI* is based on standardization with pH, SOM, IC, and clay contents of soils, which have been reported to have significant effects on the presence and mobility of PTEs in soils [13,23,35], and lithophilic elements (such as Al, Fe, and Mn) with low natural variability, as in *EF* [31]. The *SOPI* values according to average PTE content (Figure 5e) show that these soils have moderate pollution for Ni and Pb (2.55 ± 0.34 and 2.15 ± 0.15), and low pollution for Cd and Cr (1.91 ± 0.48 and 1.61 ± 0.17). In addition, the rate of contaminated soils was found to be 75.00% for Cd, 59.38% for Cr, and 9.38% for Zn. All of the soils are polluted in terms of Ni and Pb, but not in terms of Cu, Fe, and Mn (*SOPI* > 1.5).

In this study, it was found that all complex and individual index values of metals and PTEs did not have statistically significant differences (*p* > 0.05) depending on their distance from the road (Appendix A). However, with increasing traffic density, especially the complex index values decrease significantly (*p* < 0.05; Appendix A), which is another finding showing that traffic does not contribute to PTE pollution in these orchards. Interestingly, it was found that *Cf*, *ER*, *Igeo*, and *PI* values of PTEs in FC1 decreased significantly with increasing traffic density, while *EF* and *SOPI* values of Cu and Pb and Zn in FC2 increased (*p* < 0.05; Appendix A). This result shows that both *EF* and *SOPI* are appropriate individual indices for the evaluation of PTE contamination in studies conducted in areas with no local reference background values or with different soil properties.

## 4. Conclusions

By sampling soils in mixed pistachio and olive orchards along the main interurban roads in the Gaziantep region in terms of traffic density and distance from the roads, the status and origin of soil contamination with PTEs could be determined. The PTE and metal contents in the soils with high pH and IC and low SOM in Gaziantep were found to decrease in the sequence Fe > Mn > Ni > Cr > Zn > Pb > Cu > Co > Cd. The Ni contents exceeded the legal limits for agricultural soils (pH > 6) in Türkiye in 51% of the soil samples and Cr content in 18%. In the alkaline roadside soils, PTE concentrations did not differ significantly (*p* > 0.05) with respect to distance from roads, while Co, Cr, Cu, Fe, Mn, Ni, and Pb concentrations showed statistically significant differences (*p* < 0.05) with respect to traffic density. 

This study revealed that the soils of the investigated orchards are moderately contaminated with PTEs. According to *EF*, these garden soils are significantly enriched in terms of Ni and moderately enriched in terms of Cd, Cr, and Pb. According to *SOPI*, which was found to provide sensitive results in the assessment of PTE pollution in such soils with different soil general characteristics without local reference geochemical background values, Cd, Ni, and Pb contributed moderately to PTE pollution, while Cr contributed to a low degree. Therefore, the soil quality is classified as negatively changed in relation to agricultural production.

Factor analysis, explaining a large part of the total variance (84.9%), clearly revealed that pedogenic processes affect the concentrations of Co, Cr, Cu, Fe, Mn, Ni, and Pb (FC1), which have strong correlations (*r*^2^ > 0.700) with each other in these orchards, while anthropogenic pollutants contribute to the Cd and Zn (FC2) concentrations. This was similarly confirmed by correlation analyses of PTE contents and general soil properties. It is concluded that PTE pollution in these orchard soils is associated with natural processes on the one hand and the use of organic and inorganic fertilizers rather than traffic emissions on the other. 

Our study has shown that the agricultural soils in the region of Gaziantep are only slightly polluted by PTE and that road traffic does not contribute significantly. The albeit low-to-moderate pollution could be attributed to anthropogenic sources, especially regarding Cd and Zn. Therefore, against the background of a progressively increasing agricultural and industrial production, a growing threat to soils by anthropogenic pollutants must be assumed. Therefore, in the future, monitoring of PTE loads and determination of local geochemical reference background levels will be important to maintain soil health and fertility and, thus, the productivity of agricultural soils, to produce healthy food and to protect human and environmental health. 

## Figures and Tables

**Figure 1 ijerph-20-02467-f001:**
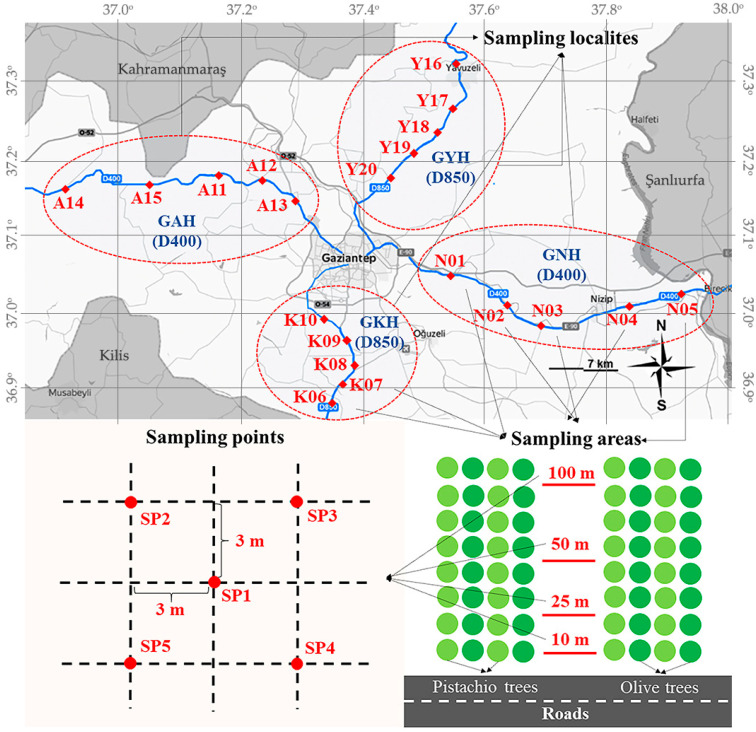
(**Top**) Map of Gaziantep (Türkiye) region, showing overland roads (in blue) along which the soil samples were taken and indicating sampling areas (in red, N01-Y20). (**Bottom right**) Sampling scheme within one sampling area and (**Bottom left**) sampling grid with sampling points (SP1–5) for composite sampling. Abbreviations are as follows; GAH, Gaziantep–Adana highway; GKH, Gaziantep–Kilis highway; GNH, Gaziantep–Nizip highway; GYH, Gaziantep–Yavuzeli highway.

**Figure 2 ijerph-20-02467-f002:**
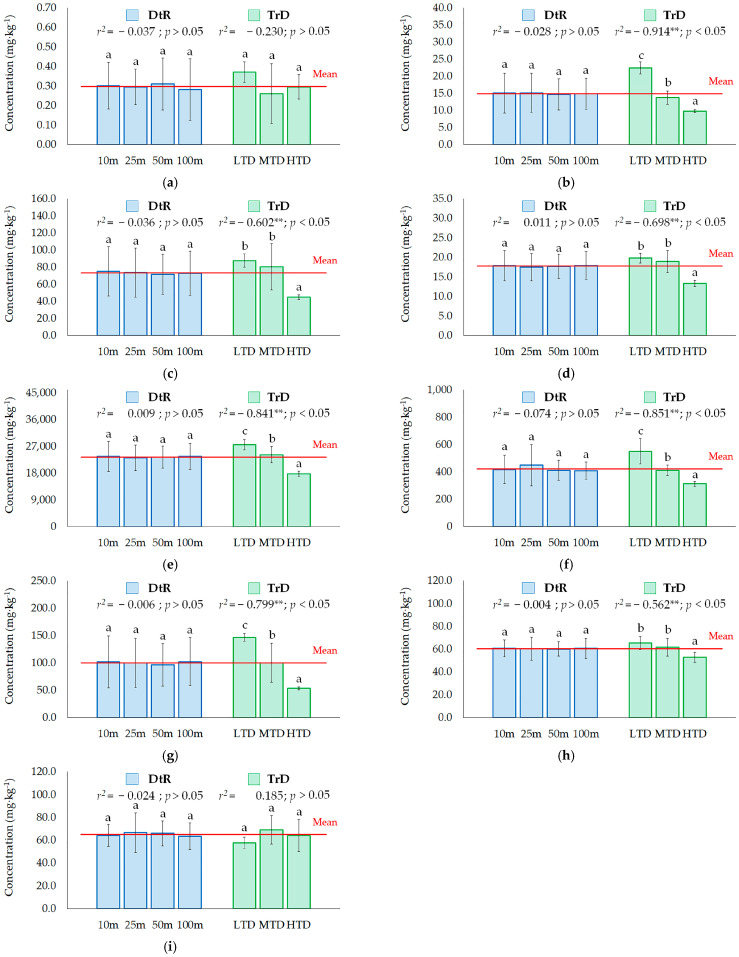
Soil concentrations (mg·kg^−1^) of Cd (**a**), Co (**b**), Cr (**c**), Cu (**d**), Fe (**e**), Mn (**f**), Ni (**g**), Pb (**h**), and Zn (**i**) (Letters a, b, and c indicate significant differences at *p* < 0.05; ** correlation is significant at the 0.01 level (2-tailed)). Abbreviations are as follows: DtR, the distance to roads; TrD, the traffic density; LTD, low traffic density; MTD, medium traffic density; HTD, high traffic density.

**Figure 3 ijerph-20-02467-f003:**
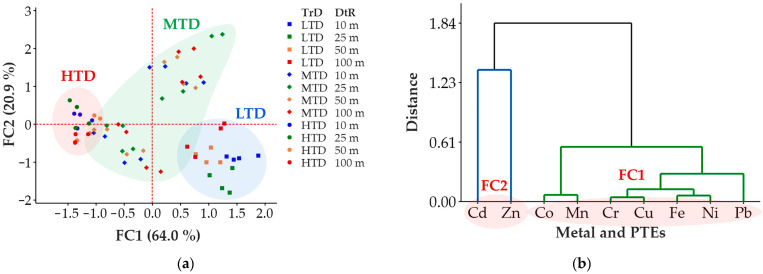
Score plots (**a**) and dendrograms (**b**) of the factor analysis of metal and PTEs’ contents in soils of orchards. Abbreviations are as follows: FC1, first factor; FC2, second factor; DtR, the distance to roads; TrD, the traffic density; LTD, low traffic density; MTD, medium traffic density; HTD, high traffic density.

**Figure 4 ijerph-20-02467-f004:**
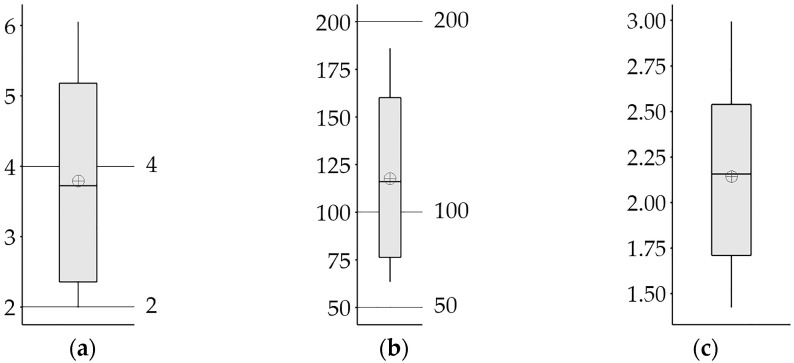
Boxplots of *mCd* (**a**), *RI* (**b**), and *PIave* (**c**) according to average values. 
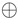
, mean value.

**Figure 5 ijerph-20-02467-f005:**
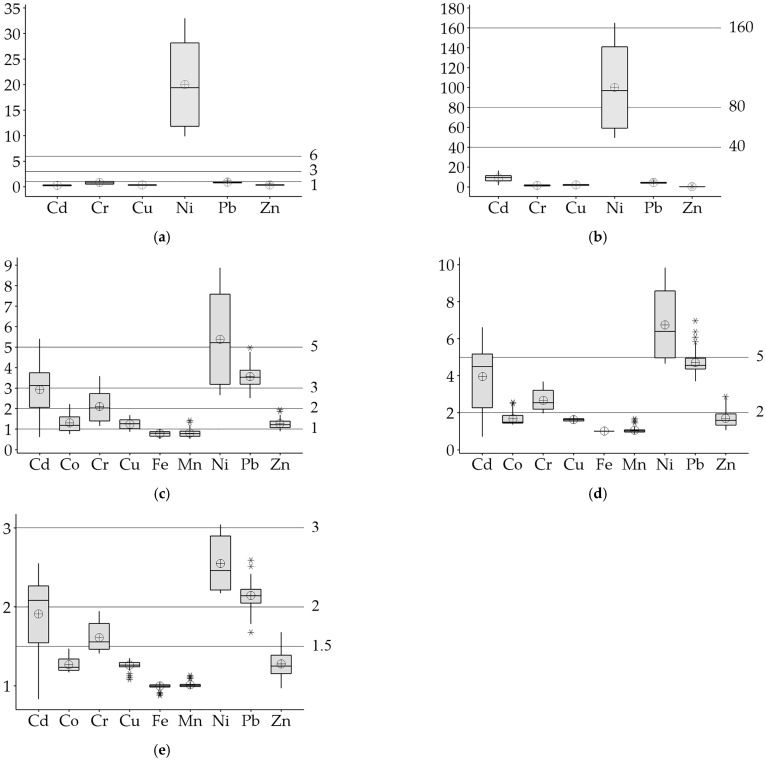
Boxplots of *Cf* (**a**), *ER* (**b**), *PI* (**c**), *EF* (**d**), and *SOPI* (**e**) according to average values. 
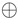
, The mean value; 
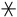
, The outlier value.

**Table 1 ijerph-20-02467-t001:** The averages and descriptive statistics of selected soil general properties in roadside orchards (Letters a, b, and c indicate significant differences at *p* < 0.05). Abbreviations are as follows; SOM, soil organic matter; IC, carbonates; LTD, low traffic density; MTD, medium traffic density; HTD, high traffic density.

			pH	SOM (%)	IC (%)	Clay (%)	Silt (%)	Sand (%)
Distance to Road	10 m (*n* = 20)	Mean	7.92 ^a^	1.08 ^a^	19.44 ^a^	23.73 ^a^	51.00 ^a^	25.27 ^a^
		SD	±0.01	±0.37	±11.66	±10.11	±8.98	±5.23
	25 m (*n* = 20)	Mean	7.92 ^a^	0.95 ^a^	20.07 ^a^	24.36 ^a^	48.13 ^a^	27.52 ^a^
		SD	±0.02	±0.52	±11.34	±9.49	±12.14	±4.63
	50 m (*n* = 20)	Mean	7.86 ^a^	0.96 ^a^	17.69 ^a^	23.36 ^a^	48.63 ^a^	28.02 ^a^
		SD	±0.06	±0.32	±12.10	±10.40	±11.30	±3.47
	100 m (*n* = 20)	Mean	7.91 ^a^	0.97 ^a^	16.07 ^a^	23.73 ^a^	47.00 ^a^	29.27 ^a^
		SD	±0.08	±0.47	±12.35	±11.22	±10.70	±5.64
		*r* ^2^	−0.237	−0.080	−0.124	−0.012	−0.125	0.296
		*p*-Value	0.191	0.730	0.909	0.998	0.900	0.423
Traffic Density	LTD (*n* = 20)	Mean	7.88 ^a^	0.92 ^ab^	4.17 ^a^	23.42 ^b^	44.63 ^a^	31.96 ^b^
		SD	±0.06	±0.15	±1.42	±3.50	±4.00	±1.41
	MTD (*n* = 40)	Mean	7.90 ^a^	1.23 ^b^	17.19 ^b^	28.97 ^b^	43.81 ^a^	27.22 ^a^
		SD	±0.06	±0.42	±3.98	±9.41	±8.24	±4.82
	HTD (*n* = 20)	Mean	7.92 ^a^	0.58 ^a^	34.70 ^c^	13.83 ^a^	62.50 ^b^	23.68 ^a^
		SD	±0.02	±0.10	±0.66	±7.09	±5.21	±3.27
		*r* ^2^	−0.294	−0.356 *	0.962 **	−0.351 *	0.617 **	−0.619 **
		*p*-Value	0.262	0.017	0.000	0.000	0.000	0.001
Total (*n* = 80)	Mean		7.90	0.99	18.31	23.79	48.69	27.52
	SD		±0.06	±0.41	±11.39	±9.82	±10.41	±4.80
	Min		7.80	0.45	2.51	8.48	34.00	18.80
	Max		8.01	1.90	35.55	39.20	68.00	37.52
	Kurt		−0.09	−0.40	−1.16	−1.04	−0.97	−0.97
	Skew		−0.37	0.82	0.26	−0.21	0.66	0.01
	CV (%)		0.70	41.36	62.20	41.27	21.38	17.46

* Correlation is significant at the 0.05 level (2-tailed); ** correlation is significant at the 0.01 level (2-tailed).

**Table 2 ijerph-20-02467-t002:** Means (mg·kg^−1^), descriptive statistics, and comparison with concentrations in different regions of the metal and PTEs detected in this study and legal limits in Türkiye. Abbreviations are as follows: Kurt, kurtosis; Skew, skewness; LGV, legal values in Türkiye; WSA, world soil average values; ESA, Europe soil average values; UCC, upper continental crust values; MAC, maximum allowable concentrations.

		Cd	Co	Cr	Cu	Fe	Mn	Ni	Pb	Zn	Ref.
Gaziantep	Mean	0.30	14.93	73.30	17.74	23,407.36	421.78	100.20	60.38	65.03	This study
	SD	±0.12	±4.94	±25.58	±3.35	±4183.76	±100.26	±41.92	±7.91	±12.19	
	Min	0.07	9.38	41.21	12.70	16,951.43	292.53	51.46	46.26	49.07	
	Max	0.53	24.73	122.66	23.57	29,891.92	725.52	159.58	82.78	98.91	
	Kurt	−0.61	−0.78	−1.45	−1.23	−1.24	1.56	−1.91	0.72	0.51	
	Skew	−0.38	0.71	0.25	−0.13	−0.17	1.10	0.06	0.62	1.00	
	CV (%)	40.87	33.11	34.90	18.89	17.87	23.77	41.83	13.10	18.74	
Bursa *		0.23	25.24	124.69	51.00		862.00	190.25	35.81	77.49	[6]
Gaziantep **		0.70	15.00	73.00	27.00			118.00	14.00	58.00	[56]
Gaziantep **						23,000.00	521.00				[57]
Malatya		0.24	12.60	59.90	36.40			70.90	14.20	67.00	[29]
Sinop				194.73	43.19	39,848.57		85.02	17.01	65.10	[12]
Şanlıurfa			16.00	85.00	27.00	37,505.00	679.00	89.00	10.60	68.00	[11]
İsfahan		0.43	14.70	85.90	35.70	28,000.00	649.90	66.20	34.60	111.40	[5]
Sulaimani ***		0.22			29.90				7.41	107.00	[13]
Peloponnese		0.54	21.99	83.12	74.68	26,500.00	1020.50	146.80	19.74	74.88	[26]
Zagreb		0.66			20.80	27,041.00	613.00	49.50	25.90	77.90	[21]
Alicante		0.34	7.10	26.50	22.50	13,608.00	295.00	20.90	22.80	52.80	[10]
Castellón		0.36	7.90	32.20	35.40	17,487.00	408.00	19.90	56.10	76.80	[25]
LGV	pH 5–6	1.00		100.00	50.00			30.00	50.00	150.00	[58]
	pH > 6	3.00		100.00	140.00			75.00	300.00	300.00	
WSA		0.06	8.00	100.00	30.00	38,000.00	600.00	40.00	10.00	50.00	[59]
ESA		0.22	13.20	111.00	23.68	28,407.00	684.00	52.00	17.79	92.00	[60]
UCC		0.10	11.60	35.00	14.30	30,890.00	527.00	18.60	17.00	52.00	[47]
MAC		5.00	50.00	200.00	150.00			60.00	300.00	300.00	[23]

* Roadside soils; ** average values of uncontaminated soil samples; *** average values of arable land.

**Table 3 ijerph-20-02467-t003:** Rotated factor loadings and communalities according to FA (extraction method, principal component analysis; rotation method, varimax with Kaiser normalization). Abbreviations are as follows: FC1, first factor; FC2, second factor; Comm, communality.

	FC1	FC2	Comm
Cd	−0.070	−0.812	0.664
Co	0.940	−0.247	0.945
Cr	0.878	0.444	0.968
Cu	0.903	0.314	0.914
Fe	0.973	0.053	0.950
Mn	0.894	−0.265	0.869
Ni	0.965	0.170	0.960
Pb	0.779	0.366	0.741
Zn	0.029	0.793	0.629
Variance	5.760	1.880	7.640
% of Var.	64.0	20.9	84.9

**Table 4 ijerph-20-02467-t004:** Correlation matrix of selected general soil properties, metals and PTE contents of the soils. Here, SOM, soil organic matters; IC, carbonates.

	Cd	Co	Cr	Cu	Fe	Mn	Ni	Pb	Zn
Co	0.117								
Cr	−0.290	0.815 **							
Cu	−0.185	0.842 **	0.962 **						
Fe	−0.026	0.923 **	0.913 **	0.937 **					
Mn	0.115	0.973 **	0.809 **	0.841 **	0.916 **				
Ni	0.078	0.970 **	0.860 **	0.887 **	0.963 **	0.943 **			
Pb	−0.201	0.780 **	0.850 **	0.845 **	0.853 **	0.741 **	0.824 **		
Zn	−0.296	0.000	0.245	0.309	0.120	−0.020	0.088	0.182	
pH	0.438 *	−0.415 *	−0.550 **	−0.553 **	−0.508 **	−0.355 *	−0.431 *	−0.540 **	−0.324
SOM	−0.358 *	0.640 **	0.772 **	0.816 **	0.697 **	0.626 **	0.685 **	0.695 **	0.561 **
IC	−0.065	−0.911 **	−0.728 **	−0.758 **	−0.857 **	−0.905 **	−0.879 **	−0.712 **	0.026
Clay	0.357 *	0.132	0.103	0.099	0.069	0.161	0.073	0.025	−0.100
Silt	−0.258	−0.428 *	−0.389 *	−0.388 *	−0.368 *	−0.482 **	−0.371 *	−0.278	0.082
Sand	−0.197	0.774 **	0.720 **	0.740 **	0.762 **	0.799 **	0.768 **	0.637 **	0.046

* Correlation is significant at the 0.05 level (2-tailed); ** correlation is significant at the 0.01 level (2-tailed).

## Data Availability

The datasets of this study are available from the corresponding author on reasonable request.

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
