# Peer review of "Status, Sources and Assessment of Potentially Toxic Element (PTE) Contamination in Roadside Orchard Soils of Gaziantep (Türkiye)"

_ijerph, 2023, doi:10.3390/ijerph20032467_

Round 1

Reviewer 1 Report

The manuscript has been reviewed. This is a nice work and enough data are supplied to support their constructive conculsions. I recommend a Minor revision after acceptance for publication.

There are some specific comments:

1. the conclusion section is too long.

2. Please use different symbol to indicate the significance between data in DtR and TrD in Figure 2.

3.Does TrD and DtR affect the soil properties?Which pose greater effects on PTEs concentration?

Author Response

Dear Reviewer

Thank you for your work on our article and for your evaluation and constructive suggestions for the revision of the article. This helped to further develop the manuscript.

1. “The conclusion section is too long”.

It has been revised

2. "Please use different symbol to indicate the significance between data in DtR and TrD in Figure 2."

Figure 2, showing the mean values of the elements with respect to DtR and TrD, is a composite of the figures arranged separately for DtR and TrD. Significance is given separately with the letters a, b, c between different distances in DtR and between different traffic densities in TrD.

3. "Does TrD and DtR affect the soil properties? Which poses greater effects on PTEs concentration ?"

The variation of soil properties due to TrD and DtR is summarized in Table 1 and further explained in L265-274. In L319-323 and L334-336, the relationships between PTE concentrations and soil properties are mentioned and the correlation results are summarized in Table 4.

In short, soil properties with DtR did not have significant differences. However, the soil properties of the roadside orchards with different traffic density had significant differences. This result reveals once again that PTE accumulation or contamination is determined by soil properties and not by traffic.

Author Response

Dear Reviewer

Thank you for your work on our article and for your evaluation and constructive suggestions for the revision of the article. This helped to further develop the manuscript.

Abstract:

1.Please mention values for metals in abstract as mentioned findings.

It’s done (L22-25).

2.conclusion should be rewritten.

It’s done (L34-35).

Introduction:

1. The authors should describe the importance of their research.

It’s done (L90-91, L93-103).

2. This part must be more focused on roadside orchard soils hence the paper needs revision.

The variation of PTE content in roadside soils (natural or agricultural) is described in L55-72.

3. why choose to study these metals?

A new sentence was added to the beginning of the introduction (L40-42) emphasizing the relevance of PTE pollution of soil. Further information on roadside contamination was added on line 53-55. Lastly, we added information at the end of the objectives (end of introduction) why we chose the selected set of metals (L107-110).

Materials & methods:

1. Clarify study area and give coordinates of specific site.

The study area is shown in detail on the map in Figure 1. We now refer to this in a new first sentence in section 2.1 (L122-126, L168).

2. Give the reason why these soil physicochemical properties were selected.

These physicochemical properties were chosen because of their effects on the presence and mobility of PTEs in soils. In the “introduction” section, the effects of soil physicochemical properties on PTE concentration and mobility are explained on L72-76). There the investigated physical-chemical soil properties are mentioned.

3. More description should be added to Figure 1 to make it more comprehensive.

It’s done.

4. Elaborate about comparation of methods for Evaluation of Environmental and Ecological Risks. It can be added in Introduction section.

We added two sentences and one reference to the beginning of section 2.3 (L176-181). This outlines, why we tested different risk indices. A longer text on a comparison of environmental and ecological risk assessment methods was included in the introduction section in an earlier version of the manuscript. We found, however, that this made the text too long and on the contrary, made the focus of the study more unclear. Instead, and based on the comments of the other reviewers, we decided to reduce the number of risk parameters tested in order to have a more straightforward selection.

5. It is suggestive to make FA a separate paragraph and Give the reason why FA was selected to identify PTE source.

We tested FA and principal component analysis (PCA) in parallel. FA and PCA gave almost the same results. The 1st and 2nd components in PCA and the 1st and 2nd factors in FA were the same. In order not to lengthen the text, only FA results are presented.  

Results and Discussion:

1. What’s the difference of “metals” and “PTEs” in the paper, why in some parts they appeared at the same time.

Explained between the first sentence and L 38-40 in the “introduction” section.  In addition, we now also distinguish between PTE and metals with regard to the selected set of analytes (L107-110).

2. Actually, the result of FA was only explained 84.9% of the total variance. PMF method is more accurate and popular.

We thank the reviewer for the constructive advice. However, the co-authors and I know probability mass function only as a tool in stochastics, informing about the probability that a discrete random variable takes on a certain value.

3. Discussion is not well elaborated, please improve. For example, explain correlation between soil physicochemical properties and metal content.

We have added additional sentences to Section 3.3 (L 322-325) that relate to the correlation between element contents and soil parameters. However, we are confident that the reasons for these correlations are familiar to professional readers. Moreover, these correlations were not an explicit research topic of this study. However, as explained in the text, they were used to show whether or not relationships exist between element contents and soil properties. In the former case, this indicates pedogenic sources, in the latter case, anthropogenic sources. Additionally, the relationships between metal contents and soil physicochemical properties are discussed together with the FA results (L322-332 and L337-339).

Conclusion:

1. Simplify the present conclusion.

The conclusion was completely revised.

2. Mention future aspects of this study. References It is suggestive to add a few latest references for the paper

It has been revised accordingly.

Reviewer 3 Report

Concentrations of Fe, Mn, Ni, Cr, Zn, Pb, Cu, Co and Cd in road side soils from Gaziantep were determined. Variations in metal concentrations, and their contamination, ecological risks were evaluated. The results are helpful for understanding the distribution and contamination level of agricultural soils of metals, and can provide useful information for environmental protection. The data were clearly presented. However, the novelty of the present work should be improved especially in the interest to broad readers.

1) Introduction part should be structured better to attract the readers' interest. The novelty of the manuscript and how it improves our understanding on the environmental fate and risk of potentially toxic element in environment should be shown.
2) L94-98:    At the end of the Introduction section, the authors address the objective of the study without presenting the rationale. The authors have to convince readers with the importance of their study. Why that area was worthy to be studied? What are new concepts that may add something to the literature?

3) L117: explain how were these soil samples collected? by what equipment? when was the sampling of soil took place? season? year? there is a lack of sampling details
4) L148: Quantification limits should be supplied.

5) L156, Table S3: the eleven indices or evaluation methods are all related to pollution assessment, so please explain the rationale of using them; can one just use the most representative index or method, which may provide the simplest and clear indication of soil pollution caused by these metal levels?

6) L265-267: provide reference(s) to support the statements.

7) L376: Geoaccumulation index and EF were used in your study and they actually reflected similar contamination status (from your description). Hence, this is not necessary to use both indices in such as case. You need to justify why you have to use both indices in the earlier text (materials and methods part of the manuscript)

8) L410: Figure 4: explain what are the symbols and shapes indicated on this graph. give further details on the caption.

9) Conclusion part should be rewritten to show what is the significance of your work for the study and to go beyond the results sections for forming the conclusions.

Author Response

Dear Reviewer

Thank you for your work on our article and for your evaluation and constructive suggestions for the revision of the article. This helped to further develop the manuscript.

1) “Introduction part should be structured better to attract the readers' interest. The novelty of the manuscript and how it improves our understanding on the environmental fate and risk of potentially toxic element in environment should be shown”.

It has been revised accordingly (L98-103).

2) “L94-98:    At the end of the Introduction section, the authors address the objective of the study without presenting the rationale. The authors have to convince readers with the importance of their study. Why that area was worthy to be studied? What are new concepts that may add something to the literature?”

It has been revised accordingly (L98-107).

3) “L117: explain how were these soil samples collected? by what equipment? when was the sampling of soil took place? season? year? there is a lack of sampling details”

Samples were taken as described in L130-134 and shown in Figure 1. Equipment, and the dates the samples were taken are included.

4) “L148: Quantification limits should be supplied”

It has been revised accordingly (Please, see at Table S2).

5) L156, Table S3: the eleven indices or evaluation methods are all related to pollution assessment, so please explain the rationale of using them; can one just use the most representative index or method, which may provide the simplest and clear indication of soil pollution caused by these metal levels?

We added two sentences and one reference to the beginning of section 2.3 (L176-178). This outlines, why we tested different risk indices. A longer text on a comparison of environmental and ecological risk assessment methods was included in the introduction section in an earlier version of the manuscript. We found, however, that this made the text too long and on the contrary, made the focus of the study more unclear. Instead, and based on the comments of the other reviewers, we decided to reduce the number of risk parameters tested in order to have a more straightforward selection.

6) “L265-267: provide reference(s) to support the statements”.

We added the reference of Kabata-Pendias to this sentence. The phrase “and possibly amplified by agricultural activities” was deleted because this statement is more speculative.

7) “L376: Geoaccumulation index and EF were used in your study and they actually reflected similar contamination status (from your description). Hence, this is not necessary to use both indices in such as case. You need to justify why you have to use both indices in the earlier text (materials and methods part of the manuscript)”

The geoaccumulation index (Igeo) is used to compare pre- and post-industrial metal concentrations, and the Enrichment factor (EF) is used to determine whether the concentration of any metal has increased in the investigated area due to natural or anthropogenic reasons. However, EF is calculated by standardizing the concentrations of the elements using the concentration of a lithophilic element in the study area. Thus, unlike Igeo, natural variations in the concentration of the element in EF were taken into account in the calculation of the index. In Igeo, the coefficient of 1.5 is used for this purpose and is not sufficient (in my opinion).

However, Geoaccumulation index skipped to reduce the number of indexes

8) L410: Figure 4: explain what are the symbols and shapes indicated on this graph. give further details on the caption.

Done as requested.

9) Conclusion part should be rewritten to show what is the significance of your work for the study and to go beyond the results sections for forming the conclusions.

Revised according to the reviewer’s comment.

Reviewer 4 Report

The manuscript (ijerph-2150984) entitled "Status, sources and assessment of potentially toxic element (PTE) contamination in roadside orchard soils of Gaziantep (Turkey)" is generally well constructed and written. The research design and data illustration are generally promising. However, before the manuscript is accepted, the following issues should be properly addressed.

(1) Metal and PTE are listed simultaneously in many places; since the definition of PTE is still controversial, it is suggested to define the PTE in the introduction.

(2) Line 36, "Science of the Total Environment, 2023,856, 158883" and "https://doi.org/10.1007/s44246-022-00010-8" could be additional up-to-date references for supporting the illustration.

(3) SOPI evaluation method is proposed firstly or from the reference? If the latter, it should be cited and introduced properly. Since the SOPI evaluation result are quite different from other classical assessment tool such as EF.

(4) Line 21, "increased" should be "decreased".

(5) Line 63, a full stop is missing.

(6) Line 84, "threat, it poses" should be revised like "threat posed".

(7) Line 275, “increased” should be “decreased”.

Author Response

Dear Reviewer

Thank you for your work on our article and for your evaluation and constructive suggestions for the revision of the article. This helped to further develop the manuscript.

(1) Metal and PTE are listed simultaneously in many places; since the definition of PTE is still controversial, it is suggested to define the PTE in the introduction.

Done as requested.

(2) Line 36, "Science of the Total Environment, 2023,856, 158883" and "https://doi.org/10.1007/s44246-022-00010-8" could be additional up-to-date references for supporting the illustration.

Done as requested (L39 and L74).

(3) “SOPI evaluation method is proposed firstly or from the reference? If the latter, it should be cited and introduced properly. Since the SOPI evaluation result are quite different from other classical assessment tool such as EF”.

The SOPI was developed in this study and suggested by us. We made this no more clear in the text of the manuscript. Hence, there is no reference for this parameter. The new index (SOPI) was created to include the different soil properties that largely influence the mobility and thus the environmental relevance of soil contamination with PTE. The four soil properties included are soil pH, and the contents of SOM, Clay and IC.

 (4) Line 21, "increased" should be "decreased".

Done as requested.

(5) Line 63, a full stop is missing.

Done as requested.

(6) Line 84, "threat, it poses" should be revised like "threat posed".

Done as requested (L90).

(7) Line 275, “increased” should be “decreased”.

Done as requested.

Round 2

Reviewer 3 Report

I recommend that the article be published in its present form.